# Association between periodontal disease due to *Campylobacter rectus* and cerebral microbleeds in acute stroke patients

Yuji Shiga[1☯], Naohisa Hosomi[2,3]*, Tomohisa Nezu[1☯], Hiromi Nishi[4], Shiro Aoki[1], Masahiro Nakamori[1,5], Kenichi Ishikawa[1,5], Naoto Kinoshita[1], Eiji Imamura[5], Hiroki Ueno[1], Tomoaki Shintani[6], Hiroki Ohge[7], Hiroyuki Kawaguchi[4], Hidemi Kurihara[8], Shinichi Wakabayashi[9], Hirofumi Maruyama[1]

1 Department of Clinical Neuroscience and Therapeutics, Hiroshima University Graduate School of Biomedical and Health Sciences, Hiroshima, Japan, 2 Department of Neurology, Chikamori Hospital, Kochi, Japan, 3 Department of Disease Model, Research Institute of Radiation Biology and Medicine, Hiroshima University, Hiroshima, Japan, 4 Department of General Dentistry, Hiroshima University Hospital, Hiroshima, Japan, 5 Department of Neurology, Suiseikai Kajikawa Hospital, Hiroshima, Japan, 6 Center of Oral Examination, Hiroshima University Hospital, Hiroshima, Japan, 7 Department of Infectious Diseases, Hiroshima University Hospital, Hiroshima, Japan, 8 Department of Periodontal Medicine, Division of Applied Life Sciences, Institute of Biomedical and Health Sciences, Hiroshima University, Hiroshima, Japan, 9 Department of Neurosurgery, Suiseikai Kajikawa Hospital, Hiroshima, Japan

☯ These authors contributed equally to this work.
* nhosomi@hiroshima-u.ac.jp

**Data Availability Statement:** All relevant data are within the paper and its Supporting Information files.

## Abstract

Oral health conditions and cerebral small vessel disease, such as white matter lesions or cerebral microbleeds (CMBs), are associated with the incidence of stroke. The purpose of this study was to examine the associations between oral health conditions (serum IgG titers of periodontal pathogens) with the presence or severity of CMBs in acute stroke patients. From January 2013 to April 2016, acute stroke patients were registered in two hospitals. Serum samples were evaluated for antibody titers against 9 periodontal pathogens using the ELISA method. The cut-off points for reactivity (the positive decision point) to each antigen were defined as more than a mean ELISA unit + 1 standard deviation (after logarithmic transformation) in all subjects. CMBs were evaluated on T2*-weighted MRI. In all, 639 patients were evaluated (ischemic, n = 533 and hemorrhagic, n = 106; 73.1 ± 12.9 years old). Among these patients, 627 were available for CMB evaluation. Among the 9 evaluated periodontal pathogens, only *Campylobacter rectus* (*C. rectus*) was associated with the presence of CMBs. the prevalence of positive serum antibody titers against *C. rectus* was higher among patients with CMBs than among those without CMBs (14.6% vs. 8.7%, P = 0.025). In addition, positive serum antibody titers against *C. rectus* remained one of the factors associated with the presence of CMBs in multivariate logistic analysis (odds ratio 2.03, 95% confidence interval 1.19–3.47, P = 0.010). A positive serum antibody titer against *C. rectus* was associated with the presence of CMBs in acute stroke patients.

**Funding:** This study was supported by research grants from the Japan Society for the Promotion 350 of Science KAKENHI (Grant Numbers 17K17350, 17K17907, 18K10746, and 351 20K16579).

**Competing interests:** Dr. Maruyama reports grants and personal fees from Eisai, grants and personal fees from Pfiser, grants and personal fees from Takeda Pharmaceutical, grants and personal fees from Otsuka Pharmaceutical, grants and personal fees from Nihon Pharmaceutical, grants and personal fees from Teijin Pharma, grants from Shionogi, grants and personal fees from Fuji Film, grants and personal fees from Boehringer Ingelheim, grants and personal fees from Sumitomo Dainippon Pharma, grants and personal fees from Nihon Medi-Physics, grants and personal fees from Bayer, grants and personal fees from MSD, grants and personal fees from Daiichi Sankyo, grants and personal fees from Kyowa Kirin, grants and personal fees from Sanofi, grants and personal fees from Novartis, grants and personal fees from Kowa Pharmaceutical, grants and personal fees from Astellas Pharma, grants and personal fees from Japan Blood Products Organization, grants and personal fees from Mitsubishi Tanabe Pharma, personal fees from Ono pharmaceutical, personal fees from Biogen, personal fees from Bristol-Myers Squibb, grants from Mylan which are unrelated to the submitted work. This does not alter our adherence to PLOS ONE policies on sharing data and materials. All other authors declare that they have no conflicts of interest.

## Introduction

Impaired oral health status, including periodontal disease, has an adverse impact on systemic health [1]. Periodontal disease associated with chronic systemic inflammation is thought to lead cardiovascular disease, stroke, and atherosclerosis progression [2]. A recent meta-analysis of cohort studies showed that the risk of ischemic or hemorrhagic stroke was significantly increased by 1.6-fold among patients with periodontitis [3]. Furthermore, an ARIC (Athero-sclerosis Risk in Communities) cohort study performed in 10362 stroke-free participants over a 15-year follow-up period showed that periodontal disease was significantly associated with the incidence of cardioembolic and thrombotic stroke [4]. They also reported that access to regular dental examinations may reduce the risk of ischemic stroke. Therefore, the management or evaluation of periodontal disease is essential when considering stroke prevention.

Although accumulating evidence indicates an association between periodontal disease and stroke, few studies have investigated whether any specific pathogen related to periodontitis is associated with the incidence of stroke. *Aggregatibacter actinomycetemcomitans* (*A. actinomycetemcomitans*) and *Porphyromonas gingivalis* (*P. gingivalis*) are classical and extensively explored "periodontal pathogens" [5]. Pussinen et al. found that systemic exposure to *P. gingivalis* but not *A. actinomycetemcomitans* was an independent risk factor for stroke in 8911 subjects [6]. Additionally, we previously reported that increased serum antibody titers against *Prevotella intermedia* (*P. intermedia*) were associated with atherothrombotic stroke patients [7]. Nishi et al. subsequently reported that increased serum antibody titers against *Fusobacterium nucleatum* (*F. nucleatum*) were associated with poor outcomes after stroke [8].

Cerebral small vessel disease (CSVD), such as cerebral microbleeds (CMBs) or white matter lesions (WMLs), are associated with the incidence of stroke [9]. The most well-established risk factor for CSVD is hypertension. Inflammation and endothelial dysfunction play key roles in the pathological cascade of CSVD [10–12]. Oral *Microbacterium* have recently been implicated in the pathogenesis of CSVD via an alternative pathway [13]. However, it is unclear whether periodontal disease or, in particular, specific pathogens that cause periodontitis, are associated with the presence or severity of CSVD. The aim of this study was to investigate whether serum IgG antibody titers against several periodontal pathogens are associated with CSVD among patients with acute stroke.

## Materials and methods

### Subjects

From January 2013 to April 2016, consecutive acute stroke patients who were categorized as ischemic or hemorrhagic were registered at Hiroshima University Hospital and Suiseikai Kajikawa Hospital. Each provided written informed consent according to a protocol approved by the Ethical Committee of Hiroshima University (Epd-614-2) and Suiseikai Kajikawa Hospital (2015–3) prior to undergoing examinations. Baseline data, including sex, age, body mass index (BMI), smoking habit, daily alcohol habit, comorbidities (hypertension, diabetes mellitus, dyslipidemia, atrial fibrillation, history of stroke and history of coronary artery disease), and serum C-reactive protein (CRP) levels (after logarithmic transformation) were collected for all patients. Imaging analysis was performed with computed tomography or magnetic resonance imaging (MRI) in all patients for the diagnosis of ischemic stroke or intracerebral hemorrhage. In the present study, patients who did not undergo MRI were excluded because the detail radiological findings for cerebral small vessel disease were evaluated by MRI. Ischemic stroke subtypes were classified using the Trial of Org 10172 in Acute

Stroke Treatment (TOAST) criteria [14] by stroke specialists. Hemorrhagic infarction and trauma-induced hemorrhage were excluded from intracerebral hemorrhage. Hypertension was defined as the use of anti-hypertensive medication before admission or a confirmed blood pressure of $\geq$140/90 mmHg at rest measured 2 weeks after onset. Diabetes mellitus was defined as a glycated hemoglobin level of $\geq$6.5%, fasting blood glucose level of $\geq$126 mg/dl, or use of anti-diabetes medication. Dyslipidemia was defined as total cholesterol level of $\geq$220 mg/dl, a low-density lipoprotein cholesterol level of $\geq$140 mg/dl, a high-density lipoprotein cholesterol level of <40 mg/dl, triglyceride levels of $\geq$150 mg/dl, or the use of anti-dyslipidemia medication. Atrial fibrillation was defined as follows: (1) a history of sustained or paroxysmal atrial fibrillation or (2) atrial fibrillation detection upon arrival or during admission. Renal function was assessed by the estimated glomerular filtration rate (eGFR) using the following revised equation for the Japanese population as follows: eGFR (ml min$^{-1}$ 1.73 m$^{-2}$) = 194 × (serum creatinine)$^{-1.094}$ × (age)$^{-0.287}$ × 0.739 (for women) [15]. Chronic kidney disease was defined as an eGFR<60 ml min$^{-1}$ 1.73 m$^{-2}$. The patient's history of stroke and coronary artery disease was collected from medical records or from the patient or the patient's family.

## Image analysis

An MRI was performed with a 1.5-T scanner (SIGNA, GE Medical Systems, Fairfield, CT, USA or Magneton Symphony Advanced or Avanto, Siemens Medical Systems, Erlangen, Germany) or a 3.0-T scanner (SIGNA, GE Medical Systems, Fairfield, CT, USA or Spectra, Siemens Healthineers, Erlangen, Germany or Philips Ingenia, Philips Medical Systems, Best, the Netherlands) within 2 weeks of the day of admission in each hospital. WMLs were evaluated on fluid-attenuated inversion recovery imaging (FLAIR), and CMBs were evaluated on T2*-weighted gradient echo imaging (T2*WI GRE). The severity of WMLs on FLAIR was assessed based on the Fazekas classification (ranging from 0 to 3) [16], with grade 0 and 2 defined as mild and grade 3 defined as severe WMLs. In this study, we defined the WMLs as periventricular hyperintensity. CMBs were defined as small-foci signal loss with a diameter of 2 to 10 mm on T2*WI. CMBs were separated from vascular flow voids, calcifications or nonhemorrhagic iron deposits. Computed tomography can facilitate the identification of suspected calcification. The locations of CMBs were classified as follows: lobar (cortex, subcortex, and white matter), deep (basal ganglia, thalamus, brain stem, and cerebellum) and mixed (both lobar and deep). WMLs and CMBs were evaluated in each hospital by two neurologists who were blind to the patients' clinical data. We examined coincidence among evaluators using 124 cases. The inter-rater agreement in WML scores (mild vs severe) and the presence of CMBs was good (Kappa coefficient = 0.85 or 0.65, respectively). In cases of disagreement, we sought consensus between the observers.

## Measurement of serum antibody titers against periodontal pathogens

Serum IgG antibody titers against periodontal pathogens were determined using enzyme-linked immunosorbent assays (ELISA) as previously described [17]. Serum samples were collected from the patients within 3 days after stroke onset and stored at −80˚C. Sonicated preparations of the following periodontal pathogens were used as bacterial antigens: *P. gingivalis* ATCC33277 (FimA type I), *A. actinomycetemcomitans* AUNY67 (Serotype c), *P. intermedia* ATCC26511, *Prevotella nigrescens (P. nigrescens)* ATCC33563, *F. nucleatum* ATCC10953, *Treponema denticola (T. denticola)* ATCC35405, *Tannerella forsythensis (T. forsythensis)* ATCC43037, *Campylobacter rectus (C. rectus)* ATCC33238, and *Eikenella corrodens (E. corrodens)* ATCC23834. We previously reported associations of these representative periodontal

pathogens with serum antibody titers and stroke outcome [17]. The serum from 5 healthy subjects was pooled and used for calibration. Using serial dilutions of the pooled control serum, the standard reaction was defined based on the ELISA unit (EU), with 100 EU corresponding to a 1:3200 dilution of the calibrator sample. The cut-off points for reactivity (the positive decision point) to each antigen were defined as more than the mean EU + 1 SD (after logarithmic transformation) obtained from all subjects in this study.

## Statistical analysis

Categorical variables are presented as numbers and percentages, and continuous variables are presented as means with standard deviations. The statistical significance of intergroup differences was assessed using $\chi^2$ tests for categorical variables and Student's t tests or Mann-Whitney U tests for continuous variables. Multivariable logistic analysis was performed with factors that were found to show a considerable difference in a univariate analysis (P < 0.2). In all analyses, P < 0.05 was considered statistically significant. All analyses were performed using JMP 14.0 (SAS Institute, Inc., Cary, NC, USA).

## Results

Among the 690 consecutive acute stroke patients, 51 who did not undergo MRI because of contraindications or unsteadiness were excluded. Thus, a total of 639 patients (73.1 ± 12.9 years old, 285 males) were enrolled in this study. The baseline clinical characteristics of these patients are presented in Table 1. There were 533 ischemic stroke patients and 106 intracerebral hemorrhage patients. The 533 ischemic stroke patients included 120 with cardioembolism, 123 with large-artery atherosclerosis, 111 with small-vessel occlusion and 179 others. The 106 intracerebral hemorrhage patients consisted of 78 hypertensives, 14 patients with cerebral amyloid angiopathy and 14 others.

### The association between serum antibody titers against periodontal pathogens and the severity of WMLs

Table 2 shows the baseline characteristics and serum antibody titers against each periodontal pathogen in patients with mild and severe WMLs. The patients with severe WMLs were significantly older, were more likely to be female, and had lower BMIs, lower rates of current smoking and daily alcohol habit, higher rates of hypertension, chronic kidney disease, and history of stroke, and higher serum CRP levels than patients with mild WMLs. In our evaluation of serum antibody titers against periodontal pathogens, there were no significant differences in the frequencies of subjects with positive titers among patient groups with different severities of WMLs.

### The association between serum antibody titers against periodontal pathogens and the presence of CMBs

Of the 639 evaluated patients, 12 were excluded because of artifacts caused by body movement during the evaluation of CMBs on T2*WI. Hence, 627 patients were analyzed (ischemic, n = 522 and hemorrhagic, n = 105). The clinical characteristics of patients with and without CMBs are shown in Table 3. The patients with CMBs were significantly older and had higher frequencies of hypertension and history of stroke than were found in those without CMBs. Of the 9 periodontal pathogens for which we evaluated serum antibody titers, we found that only serum antibody titers against *C. rectus* were associated with the presence of CMBs. The frequency of positive serum antibody titers against *C. rectus* was higher among patients with

**Table 1. Baseline characteristics.**

|  | n = 639 |
|---|---|
| Age (years) | 73.1 ± 12.9 |
| Female, n (%) | 285 (44.6) |
| Body mass index, kg/m$^2$ | 22.8 ± 4.1 |
| Hypertension, n (%) | 487 (76.2) |
| Diabetes mellitus, n (%) | 166 (26.0) |
| Dyslipidemia, n (%) | 258 (40.4) |
| Atrial fibrillation, n (%) | 121 (18.9) |
| Chronic kidney disease, n (%) | 214 (33.5) |
| History of stroke, n (%) | 209 (32.7) |
| History of coronary artery disease, n (%) | 51 (8.0) |
| Current Smoker (n = 621), n (%) | 126 (19.9) |
| Daily alcohol habit (n = 619), n (%) | 137 (21.7) |
| **Ischemic Stroke (n = 533)** |  |
| Cardioembolism, n (%) | 120 (22.5) |
| Large-artery atherosclerosis, n (%) | 123 (23.1) |
| Small-vessel occlusion, n (%) | 111 (20.8) |
| Others, n (%) | 179 (33.6) |
| **Intracerebral hemorrhage (n = 106)** |  |
| Hypertensive, n (%) | 78 (73.6) |
| Cerebral amyloid angiopathy, n (%) | 14 (13.2) |
| Others, n (%) | 14 (13.2) |
| **MRI findings** |  |
| White matter lesions (WMLs) |  |
| Fazekas Grade 0, n (%) | 39 (6.1) |
| Fazekas Grade 1, n (%) | 167 (26.1) |
| Fazekas Grade 2, n (%) | 236 (36.9) |
| Fazekas Grade 3, n (%) | 197 (30.8) |
| Presence of CMBs (n = 627), n (%) | 315 (50.2) |
| Multiple CMBs (n = 627), n (%) | 215 (34.3) |

Data are presented as the means ± SD for age and body mass index; and as the number of patients (%) for other parameters. MRI, magnetic resonance imaging; CMBs, cerebral microbleeds.

CMBs than those without CMBs (14.6% vs. 8.7%, P = 0.025). Multivariate logistic analysis revealed that a positive serum antibody titer against *C. rectus* was associated with the presence of CMBs after adjustment for factors that showed considerable differences (odds ratio 2.04, 95% confidence interval 1.19–3.48, P = 0.010) (Table 4). CRP levels were not associated with serum antibody titers against *C. rectus* (ρ = 0.019, p = 0.64, n = 627; S1 Fig).

## The association between *C. rectus* and the number or locations of CMBs

Although the difference was not significant, the total number of CMBs was slightly higher in patients with a positive antibody titer against *C. rectus* than in those without (median [interquartile range], 1 [0–4] vs. 0 [0–3], P = 0.077) (Fig 1). A positive serum antibody titer against *C. rectus* was not associated with multiple CMBs (more than 2) in either the univariate or multivariate logistic analysis. The patients were divided into four categories according to the location of CMBs (none, deep, lobar, and mixed). In all, 27 of the 312 patients (8.7%) with no

**Table 2. Baseline characteristics and serum IgG titers against periodontal pathogens of patients with mild and severer white matter lesions (WMLs).**

| | mild WMLs (0–2) n = 442 | severe WMLs (3) n = 197 | p |
|---|---|---|---|
| Age (years) | 70.0 ± 13.0 | 80.0 ± 9.6 | < .001* |
| Female, n (%) | 180 (40.7) | 105 (53.3) | 0.003* |
| Body mass index, kg/m$^2$ | 23.2 ± 4.1 | 21.8 ± 3.7 | < .001* |
| Hypertension, n (%) | 325 (73.5) | 162 (82.2) | 0.020* |
| Diabetes mellitus, n (%) | 112 (25.3) | 54 (27.4) | 0.625 |
| Dyslipidemia, n (%) | 185 (41.9) | 73 (37.1) | 0.258 |
| Atrial fibrillation, n (%) | 75 (17.0) | 46 (23.4) | 0.063 |
| Chronic kidney disease, n (%) | 126 (28.5) | 88 (44.7) | < .001* |
| History of stroke, n (%) | 108 (24.4) | 101 (51.3) | < .001* |
| History of coronary artery disease, n (%) | 31 (7.0) | 20 (10.2) | 0.206 |
| Current Smoking (n = 633), n (%) | 106 (24.3) (n = 436) | 20 (10.2) (n = 197) | < .001* |
| Daily alcohol habit, n (%) | 117 (26.9) (n = 435) | 20 (10.2) (n = 196) | < .001* |
| Serum CRP, log mg/dl | -0.82 ± 0.70 | -0.68 ± 0.72 | 0.007* |
| **Frequencies of positivity for periodontal pathogens** | | | |
| *P. gingivalis* ATCC33277 (FimA type I), n (%) | 56 (12.7) | 24 (12.2) | 0.898 |
| *A. actinomycetemcomitans* AUNY67 (Serotype c), n (%) | 55 (12.4) | 18 (9.1) | 0.281 |
| *P. intermedia* ATCC26511, n (%) | 52 (11.8) | 27 (13.7) | 0.516 |
| *P. nigrescens* ATCC33563, n (%) | 49 (11.1) | 19 (9.6) | 0.677 |
| *F. nucleatum* ATCC10953, n (%) | 53 (12.0) | 34 (17.3) | 0.081 |
| *T. denticola* ATCC35405, n (%) | 71 (16.1) | 30 (15.2) | 0.816 |
| *T. forsythensis* ATCC43037, n (%) | 58 (13.1) | 30 (15.2) | 0.534 |
| *C. rectus* ATCC33238, n (%) | 47 (10.6) | 26 (13.2) | 0.349 |
| *E. corrodens* ATCC23834, n (%) | 58 (13.1) | 23 (11.7) | 0.700 |

Data are presented as means ± SD for age, body mass index, serum CRP and as the number of patients (%) for other parameters. CRP, C-reactive protein.

*P < 0.05 (statistically significant).

CMBs, 19 of the 118 patients (16.1%) with deep CMBs, 10 of the 45 patients (22.2%) with lobar CMBs, and 17 of the 152 patients (11.2%) with mixed CMBs were positive for serum antibody titers against *C. rectus* (P = 0.019) (Fig 2). These distributions were also similar after the patients were divided into those with ischemic stroke and those with intracerebral hemorrhage (S2 and S3 Figs).

## Discussion

This study demonstrates that a positive serum IgG antibody titer against *C. rectus* is associated with the presence of CMBs in acute stroke patients after adjusting for baseline covariate factors.

CSVDs, including lacunar infarction, WMLs, and CMBs, are common phenomena associated with ageing that worsen with hypertension and diabetes mellitus [18]. It has been suggested that chronic inflammation and blood-brain barrier disruption are associated with CSVDs [19]. Several studies have demonstrated an association between lacunar infarction and chronic periodontal disease, which is thought to induce chronic inflammation [20, 21]. Chronic periodontal disease has also been independently associated with the presence of lacunar infarction after adjustment for several vascular risk factors [20]. In addition, the severity of periodontitis tended to be associated with the number of lacunar infarctions. The progression of periodontitis may lead to an increase in the number of lacunar infarctions [21]. Whether

**Table 3. Baseline characteristics and serum IgG titers against periodontal pathogens of patients with and without cerebral microbleeds (CMBs).**

| | Patients without CMBs n = 312 | Patients with CMBs n = 315 | p |
|---|---|---|---|
| Age (years) | 71.7 ± 13.4 | 74.6 ± 12.2 | 0.008* |
| Female, n (%) | 148 (47.4) | 132 (41.9) | 0.173 |
| Body mass index, kg/m² | 23.1 ± 4.1 | 22.5 ± 4.0 | 0.091* |
| Hypertension, n (%) | 214 (68.6) | 267 (84.8) | < .001* |
| Diabetes mellitus, n (%) | 74 (23.7) | 88 (27.9) | 0.237 |
| Dyslipidemia, n (%) | 134 (43.0) | 120 (38.1) | 0.223 |
| Atrial fibrillation, n (%) | 68 (21.8) | 50 (15.9) | 0.066 |
| Chronic kidney disease, n (%) | 93 (29.8) | 118 (37.5) | 0.052 |
| History of stroke, n (%) | 82 (26.3) | 123 (39.1) | < .001* |
| History of coronary artery disease, n (%) | 24 (7.7) | 26 (8.3) | 0.883 |
| Current smoking, n (%) | 68 (21.9) (n = 310) | 55 (17.7) (n = 311) | 0.192 |
| Daily alcohol habit, n (%) | 69 (22.3) (n = 310) | 64 (20.7) (n = 309) | 0.696 |
| Serum CRP, log mg/dl | -0.81 ± 0.73 | -0.76 ± 0.69 | 0.191 |
| **Frequencies of positivity for periodontal pathogens** | | | |
| *P. gingivalis* ATCC33277 (FimA type I), n (%) | 36 (11.5) | 44 (14.0) | 0.403 |
| *A. actinomycetemcomitans* AUNY67 (Serotype c), n (%) | 40 (12.8) | 32 (10.2) | 0.318 |
| *P. intermedia* ATCC26511, n (%) | 35 (11.2) | 44 (14.0) | 0.336 |
| *P. nigrescens* ATCC33563, n (%) | 30 (9.6) | 37 (11.8) | 0.439 |
| *F. nucleatum* ATCC10953, n (%) | 42 (13.5) | 45 (14.3) | 0.818 |
| *T. denticola* ATCC35405, n (%) | 49 (15.7) | 52 (16.5) | 0.828 |
| *T. forsythensis* ATCC43037, n (%) | 43 (13.8) | 45 (14.3) | 0.909 |
| *C. rectus* ATCC33238, n (%) | 27 (8.7) | 46 (14.6) | 0.025* |
| *E. corrodens* ATCC23834, n (%) | 38 (12.2) | 42 (12.8) | 0.720 |

Data are presented as means ± SD for age, body mass index, serum CRP and as the number of patients (%) for other parameters. CRP, C-reactive protein.

*P < 0.05 (statistically significant).

**Table 4. Indicators associated with the presence of CMBs.**

| | Odds ratio | 95%CI | p-value |
|---|---|---|---|
| Age | 1.02 | 1.00–1.03 | 0.047* |
| Female | 0.63 | 0.43–0.90 | 0.013* |
| BMI | 0.96 | 0.91–1.00 | 0.054 |
| Hypertension | 2.66 | 1.75–4.03 | <0.001* |
| Atrial fibrillation | 0.52 | 0.33–0.82 | 0.005* |
| Chronic kidney disease | 1.16 | 0.80–1.67 | 0.436 |
| History of stroke | 1.56 | 1.09–2.24 | 0.016* |
| Current smoking | 0.82 | 0.53–1.29 | 0.401 |
| Serum CRP | 1.12 | 0.88–1.41 | 0.356 |
| *C. rectus* positive | 2.04 | 1.19–3.48 | 0.010* |

Multivariate logistic analysis was performed with the factors in Table 3 that showed considerable differences in univariate analysis (p < 0.2) for the presence of CMBs. *C. rectus*, indicates *Campylobacter rectus*; CMBs, cerebral microbleeds; CI, confidence interval; BMI, body mass index; CRP, C-reactive protein.

*P < 0.05 (statistically significant).

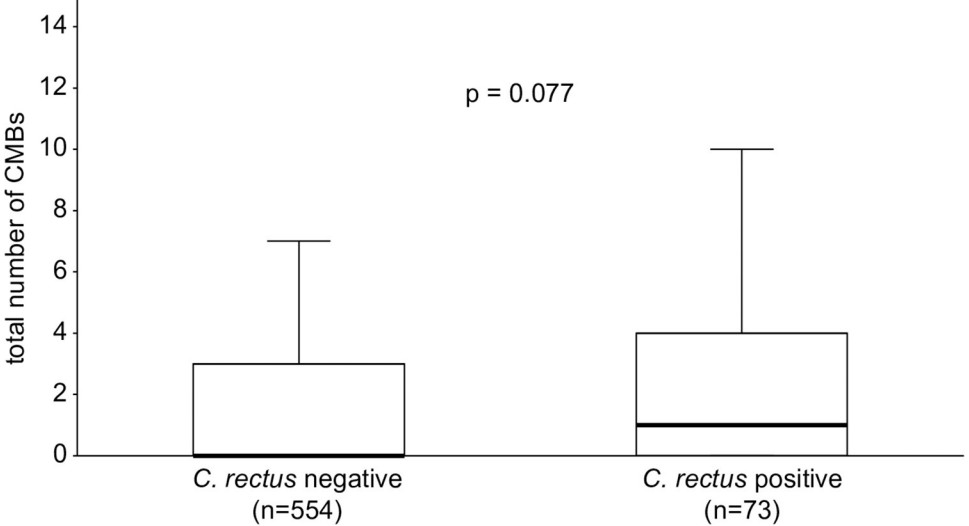

**Fig 1. Association between the total number of CMBs and positive serum antibody titers against *C. rectus*.** The total number of CMBs was non-significantly higher in patients with positive serum antibody titers against *C. rectus* than in patients with negative tests (median (IQR), 1 (0–4) vs. 0 (0–3), P = 0.077). The error bars show the local minimum (10%) and local maximum (90%) after excluding outliers. *C. rectus*: *Campylobacter rectus*; CMBs: cerebral microbleeds.

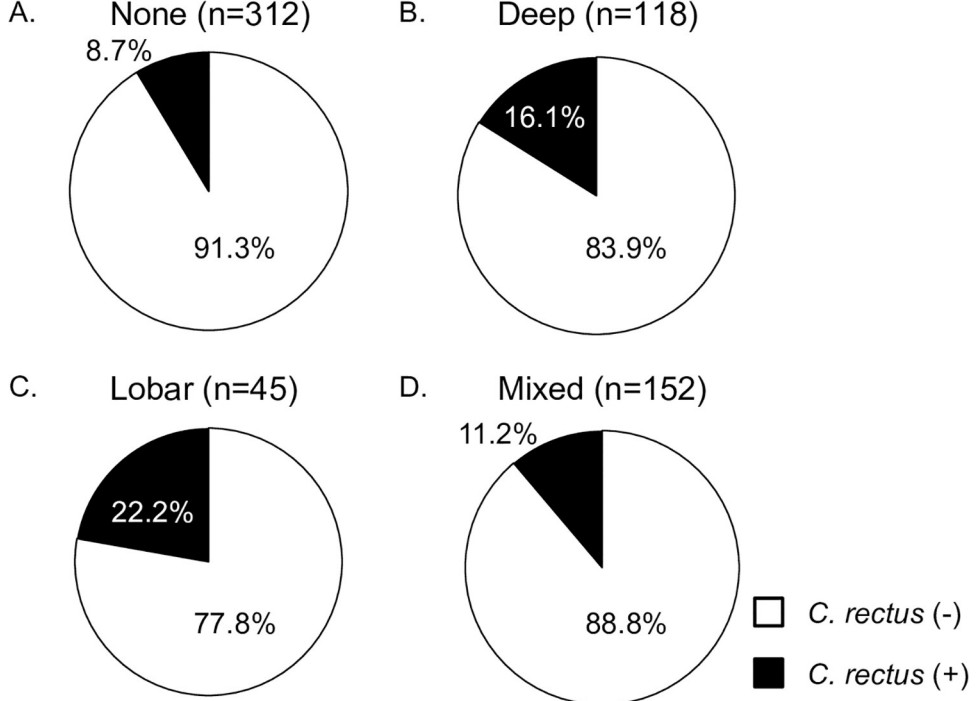

**Fig 2. Association between the locations of CMBs and positive serum antibody titers against *C. rectus*.** *C. rectus* was detected in 16.1% of patients with deep CMBs, in 22.2% of those with lobar CMBs, in 8.7% of those with no CMBs and in 11.2% of those with mixed CMBs. *C. rectus*: *Campylobacter rectus*; CMBs: cerebral microbleeds.

periodontal disease is associated with the severity of WMLs has not been determined. Tooth loss caused by the progression of periodontal disease is associated with the severity of WMLs [22, 23]. However, in our study, we did not find that any specific periodontal disease antibody titer was associated with the severity of WMLs.

Of the 9 antibodies against periodontal pathogens for which we checked serum titer levels, a positive serum antibody titer for *C. rectus* was associated with the presence of CMBs. When we considered the locations of CMBs, we found that the frequency of patients with a positive serum antibody titer against *C. rectus* was high in both patients with deep and patients with lobar CMBs. In addition, patients with a positive serum antibody titer against *C. rectus* did not have higher numbers of CMBs than were found in those without. These results may indicate that a positive serum antibody titer against *C. rectus* is associated with the initial occurrence of CMBs but not with an increase in the number of CMBs. In general, deep CMBs were associated with hypertension, and lobar CMBs were associated with amyloid angiopathy [24]. Hypertension was the most important factor that was related to the number of CMBs [25]. Indeed, hypertension was closely associated with the presence of CMBs in the present study. In recent years, it has been suggested that *cnm* gene-positive *Streptococcus mutans* (*cnm*-positive *S. mutans*) is related to the occurrence of CMBs in deep regions. Bacteria residing in the oral cavity may be a source of CMBs, which may be induced via different pathways in these patients than has been observed in hypertension and cerebral amyloid angiopathy [26]. Therefore, the association between oral bacteria and CMBs has received considerable attention. Our results indicate that a positive serum antibody titer against *C. rectus* is associated with the presence of CMBs and provide further evidence supporting the notion that oral health conditions play a role in the occurrence of CMBs.

*C. rectus* is a gram-negative, microaerophilic, and motile bacterium associated with periodontal disease [27, 28]. *C. rectus* is also thought to play a pathogenic role in human disease. It has been reported that the prevalence of periodontal pathogens, including *C. rectus*, is higher in the subgingival biofilms of patients with coronary artery disease than in non-cardiac subjects. Furthermore, *C. rectus* and other periodontal pathogens were significantly associated between the subgingival and atherosclerotic plaques in cardiac patients [29]. Although some case reports have shown that there is an association between *C. rectus* and cavernous sinus thrombosis [30, 31], no cohort study has investigated the association between *C. rectus* and neurological disease. However, Misaki et al. have reported that the coexistence of *C. rectus* with *cnm*-positive *S. mutans* in saliva was associated with high urinary protein levels in patients with immunoglobulin A nephropathy (IgAN) [32]. Microalbumin reflects endothelial dysfunction, and endothelial dysfunction is thought to play a role in the mechanisms leading to CSVD-related brain changes [33]. Therefore, *C. rectus* and *cnm*-positive *S. mutans* may be associated with systemic small vessel disease outbreaks. Future studies must therefore examine whether the coexistence of *C. rectus* with *cnm*-positive *S. mutans* is related to CMB occurrence.

This study has several limitations. First, we could not evaluate the clinical periodontal examination of all patients. It is not fully clear how serum antibody titers against periodontal disease are influenced by the presence or severity of periodontal disease. Second, this study targeted acute ischemic stroke patients, and various factors, such as systemic inflammation involving CRP or several cytokines, might therefore affect the association between serum antibody titers and both periodontal disease and CSVD [34, 35]. Although we could not evaluate cytokine-levels, we found no significant associations between serum antibody titers and CRP levels (S1 Table). Hence, the association between serum periodontal antibody titers and CSVD is likely not due to systemic inflammation alone. Third, there may be a lot of variability in the healthy controls due to the small number included (5 healthy subjects). This might lead to potential concerns about the statistical analysis of the serum antibody data. Fourth, we did not focus on

increased levels of titers of each periodontal pathogen, although we focused on the positive rate. In fact, there is no significant difference in the association with serum IgG titer against *C. rectus* and the presence of CMBs (S2 Table). The limitation of this study was that we revealed only the association with positive rate of serum IgG titers and presence of CMBs. Finally, because this is a cross-sectional study, the causal relationship between the progression of periodontal diseases and the occurrence of CSVD was not explored. Further studies should investigate the association between serum antibody titers and both periodontal disease and CSVD in the general population.

In conclusion, serum IgG antibody titers against *C. rectus* are high in acute stroke patients with CMBs. Further clinical and basic studies should be performed to determine the influence of *C. rectus* on the pathophysiological status of CSVD.

## Supporting information

**S1 Fig. Correlation analyses between serum antibody titers against *C. rectus* and serum CRP levels.** The analyses were performed using Spearman's rank correlation coefficient. (TIF)

**S2 Fig. Associations between the locations of CMBs and positive serum antibody titers against *C. rectus*. in ischemic stroke patients.** (TIF)

**S3 Fig. Associations between the locations of CMBs and positive serum antibody titers against *C. rectus*. in intracerebral hemorrhage patients.** (TIF)

**S1 Table. Correlation analyses between serum antibody titers against 9 periodontal pathogens and serum CRP levels.** The analyses were performed using Spearman's rank correlation coefficient. (TIF)

**S2 Table. Associations between the presence of CMBs and serum IgG titer against periodontal pathogens.** (TIF)

**S1 Dataset.** (XLSX)

## Author Contributions

**Data curation:** Yuji Shiga, Tomohisa Nezu, Shiro Aoki, Masahiro Nakamori, Kenichi Ishikawa, Naoto Kinoshita, Eiji Imamura, Hiroki Ueno, Shinichi Wakabayashi.

**Formal analysis:** Yuji Shiga, Tomohisa Nezu.

**Investigation:** Yuji Shiga, Hiromi Nishi, Tomoaki Shintani, Hiroyuki Kawaguchi.

**Methodology:** Hiromi Nishi, Shiro Aoki, Tomoaki Shintani, Hiroki Ohge, Hiroyuki Kawaguchi, Hidemi Kurihara.

**Supervision:** Naohisa Hosomi, Hiroki Ohge, Hidemi Kurihara, Hirofumi Maruyama.

**Writing – original draft:** Yuji Shiga, Tomohisa Nezu.

**Writing – review & editing:** Naohisa Hosomi, Tomohisa Nezu, Hirofumi Maruyama.

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
