## [Decision Letter · Decision Letter 0]

30 Jul 2020

PONE-D-20-11864

Association between periodontal disease due to Campylobacter rectus and cerebral microbleeds in acute stroke patients

PLOS ONE

Dear Dr. Hosomi,

Thank you for submitting your manuscript to PLOS ONE. After careful consideration, we feel that it has merit but does not fully meet PLOS ONE’s publication criteria as it currently stands. Therefore, we invite you to submit a revised version of the manuscript that addresses the points raised during the review process.

The manuscript would be enhanced with the inclusion of a range of T2*-weighted MRI for white matter lesion and cerebral microbleeds. The authors should address the correlation between the severity of lesions in MRI and periodontal pathogens.  

We look forward to receiving your revised manuscript.

Kind regards,

Hansel McClear Fletcher, Ph.D.

Academic Editor

PLOS ONE

Journal Requirements:

2. Thank you for including your competing interests statement;

"Dr. Maruyama reports grants and personal fees from Eisai, grants and personal fees from Pfiser, grants and personal fees from Takeda Pharmaceutical, grants and personal fees from Otsuka Pharmaceutical, grants and personal fees from Nihon Pharmaceutical, grants and personal fees from Teijin Pharma, grants from Shionogi, grants and personal fees from Fuji Film, grants and personal fees from Boehringer Ingelheim, grants and personal fees from Sumitomo Dainippon Pharma, grants and personal fees from Nihon Medi-Physics, grants and personal fees from Bayer, grants and personal fees from MSD, grants and personal fees from Daiichi Sankyo, grants and personal fees from Kyowa Kirin, grants and personal fees from Sanofi, grants and personal fees from Novartis, grants and personal fees from Kowa Pharmaceutical , grants and personal fees from Astellas Pharma, grants and personal fees from Japan Blood Products Organization, grants and personal fees from Mitsubishi Tanabe Pharma, personal fees from Ono pharmaceutical, personal fees from Biogen, personal fees from Bristol-Myers Squibb, grants from Mylan, outside the submitted work. All other authors declare that they have no conflicts of interest."

Reviewers' comments:

Reviewer's Responses to Questions

**Comments to the Author**

1. Is the manuscript technically sound, and do the data support the conclusions?

Reviewer #1: Partly

Reviewer #2: Yes

2. Has the statistical analysis been performed appropriately and rigorously? 

Reviewer #1: I Don't Know

Reviewer #2: Yes

3. Have the authors made all data underlying the findings in their manuscript fully available?

Reviewer #1: No

Reviewer #2: No

4. Is the manuscript presented in an intelligible fashion and written in standard English?

Reviewer #1: Yes

Reviewer #2: Yes

5. Review Comments to the Author

Reviewer #1: The submission by Shiga et al. describes immunodetection of C. rectus at elevated levels in patients with cerebral microbleeds in acute stroke patients. The strengths of the manuscript are in the organization and presentation of data in the manuscript, a study that directly examines associations using human clinical data, and identification of C. rectus as on oral organism associated with microbleed in acute stroke patients. Areas of weakness include the lack of oral examination at the time of patient evaluation/serum collection, the lack of oral subgingival biofilm samples to identify the presence/absence of C. rectus in these oral samples, and reliance on antibody levels (ELISA) data only to indicate an association. Despite the clear omissions noted, this reviewer is weakly enthusiastic about the manuscript. Specific comments are as follows:

1- As identified by the authors, the strongest associated factor in acute stroke patients was hypertension and this aligns with the disease as an anticipated result. C. rectus is a significant factor in this study, but other periodontal organisms such as P. gingivalis, and A. actinomycetemcomitans have been associated with stroke previously in a study that had substantially more patients than the current study. What would be the precise limitation of enrolling fewer patients and observing differences in reported outcomes between the present study and the Pussinen et al. study that is referenced?

2- In the Methods, the number of patients should be identified, also, there is no mention of the numbers of total patients approached for exclusion in this study, or if there are any exclusion criteria.

3- There is essentially no description of the ELISA assay, determination of what type of antibodies (all, IgM, IgG, etc.) are being screed for in the sera. Further, it is not clear why pooled serum from only 5 health individuals was used to control? A similar number of control subject sera should be included in the study data, and there should be no pooling of samples as it is known that many periodontally healthy individual can have relatively different levels of circulation serum antibodies to periodontal disease-associated bacteria. This leads to potential concerns about statistical analysis of these serum antibody data. There is no description of the specific strains of each bacteria used in the present study. This must be included as it is known that organisms such as P. gingivalis have a fairly high degree of heterogeneity.

4- Page 14 (lines 209-211). These data are different than what is presented on page 10. Please clarify.

5- Page 20 (lines 291-294). With the knowledge of S. mutans associations association with CMBs, it is curious that in the present manuscript that S. mutans levels of specific antibody were not assessed in the patients examined? Why only the focus on the subgingival group? Indeed, this reviewer thinks that having these data would make for an interesting and important comparison to prior work in this area.

6- CRP was the only inflammatory marker examined. Numerous studies support that pro-inflammatory cytokines serve as important markers of overall systemic inflammation. Further, systemic inflammation is liked to IL-1 (Sobowale et al. Stroke 2016. 47:2160-67). Evaluation of the levels of other inflammatory markers would improve understanding of potential underlying mechanisms in this association, and are thus suggested.

Reviewer #2: The authors aim to evaluate the associations between periodontal pathogens with the presence or severity of cerebral microbleeds and white matter lesions in acute stroke patients. There are good numbers of observational studies that periodontal disease is highly associated with the incidence of cardioembolic and thrombotic stroke. Understanding the whether any specific types of pathogen related to periodontitis is associated with the incidence of stroke is highly crucial.

The experimental design is well organized, and the results might be impactful to the field of stroke and cerebrovascular disorders.

I have only one minor suggestion.

It is a well written and taught study, which would be impactful in the field. However, the quality/number of the figures are quite low, I will suggest improving numbers of figures including a range of descriptive T2*-weighted MRI for white matter lesion and cerebral microbleeds. It would necessary if authors have correlative illustrations between severity of lesions in MRI and of periodontal pathogens. This figure would be increase quality of presentation.

6. PLOS authors have the option to publish the peer review history of their article (what does this mean?). If published, this will include your full peer review and any attached files.

Reviewer #1: No

Reviewer #2: No

---

## [Author Response · Author response to Decision Letter 0]

7 Aug 2020

Thank you again for reviewing our manuscript. We appreciate the insightful comments and advice of the reviewers. We have provided point-by-point responses to each of the comments and highlighted the corresponding revisions in the Respond to Reviewers Word document.

---

## [Decision Letter · Decision Letter 1]

2 Sep 2020

PONE-D-20-11864R1

Association between periodontal disease due to Campylobacter rectus and cerebral microbleeds in acute stroke patients

PLOS ONE

Dear Dr. Hosomi,

Thank you for submitting your manuscript to PLOS ONE. After careful consideration, we feel that it has merit but does not fully meet PLOS ONE’s publication criteria as it currently stands. Therefore, we invite you to submit a revised version of the manuscript that addresses the points raised during the review process.

Please clearly list the limitations of the study and clarify author contributions.

We look forward to receiving your revised manuscript.

Kind regards,

Hansel McClear Fletcher, Ph.D.

Academic Editor

PLOS ONE

Reviewers' comments:

Reviewer's Responses to Questions

**Comments to the Author**

1. If the authors have adequately addressed your comments raised in a previous round of review and you feel that this manuscript is now acceptable for publication, you may indicate that here to bypass the “Comments to the Author” section, enter your conflict of interest statement in the “Confidential to Editor” section, and submit your "Accept" recommendation.

Reviewer #1: (No Response)

Reviewer #2: All comments have been addressed

2. Is the manuscript technically sound, and do the data support the conclusions?

Reviewer #1: Partly

Reviewer #2: Yes

3. Has the statistical analysis been performed appropriately and rigorously? 

Reviewer #1: I Don't Know

Reviewer #2: Yes

4. Have the authors made all data underlying the findings in their manuscript fully available?

Reviewer #1: No

Reviewer #2: Yes

5. Is the manuscript presented in an intelligible fashion and written in standard English?

Reviewer #1: Yes

Reviewer #2: Yes

6. Review Comments to the Author

Reviewer #1: The revised submission PONE-D-20-11864R1 has made significant improvements from the interesting original submission. As few items are felt to require additional attention.

1- In the limitations of the study, the lack of a clinical periodontal exam of all patients included in this study should be listed as limitation #1 in the listing of limitations.

2- In the limitations of the study, the lack of a similar number of controls, and the handling of the 5 samples (as pooled) should be moved to limitation #2 in the listing of limitations described in the discussion.

3- This reviewer thanks the authors for identifying that IgG levels were determined. However, the indication of “titer” is not presented in any of the tables, rather “% positive” was used to compare between clinical groups. This reviewer suggests that the serum titer +/- error for each bacterium for each of the patient/serum samples examined be included in each of the tables reporting titer information.

Minor points:

1- Please define “WMLs” on first use in the text.

2- This reviewer thanks the authors for including the strain examination data for the various bacterial lysates in the response to reviewer comments. Please apply the strain information throughout the Methods, and presented data (all relevant Tables).

3- Table 2 and Table 3 titles. Please change “antibody” to “IgG”.

4- In all tables, please identify using symbols, those data that demonstrate significant differences between groups. Please identify and describe the symbol and test used in each of the corresponding figure legends, table descriptions, respectively.

Reviewer #2: (No Response)

7. PLOS authors have the option to publish the peer review history of their article (what does this mean?). If published, this will include your full peer review and any attached files.

Reviewer #1: No

Reviewer #2: **Yes: **Onder Albayram

---

## [Author Response · Author response to Decision Letter 1]

7 Sep 2020

Thank you for your comments. We have tried to respond to your comments as much as possible. We have highlighted the points of correction using red color letters in the revised text. We hope that our response meets your requests.

---

## [Editor Report · Decision Letter 2]

14 Sep 2020

Association between periodontal disease due to Campylobacter rectus and cerebral microbleeds in acute stroke patients

PONE-D-20-11864R2

Dear Dr. Hosomi,

We’re pleased to inform you that your manuscript has been judged scientifically suitable for publication and will be formally accepted for publication once it meets all outstanding technical requirements.

Kind regards,

Hansel McClear Fletcher, Ph.D.

Academic Editor

PLOS ONE

---

## [Editor Report · Acceptance letter]

29 Sep 2020

PONE-D-20-11864R2 

Association between periodontal disease due to *Campylobacter rectus* and cerebral microbleeds in acute stroke patients 

Dear Dr. Hosomi:

I'm pleased to inform you that your manuscript has been deemed suitable for publication in PLOS ONE. Congratulations! Your manuscript is now with our production department. 

Kind regards, 

on behalf of

Dr. Hansel McClear Fletcher 

Academic Editor

PLOS ONE